# Evaluation of *Aspergillus flavus* Growth on Weathered HDPE Plastics Contaminated with Diesel Fuel

**DOI:** 10.3390/microorganisms13061418

**Published:** 2025-06-18

**Authors:** Juan Valenzuela, César Sáez-Navarrete, Xavier Baraza, Fernando Martínez, Bastián Márquez

**Affiliations:** 1Department of Chemical and Bioprocess Engineering, School of Engineering, Pontificia Universidad Católica de Chile, Av. Vicuña Mackenna 4860, Macul, Santiago 7820436, Chile; csaezn@uc.cl (C.S.-N.); basty.marquez.g@uc.cl (B.M.); 2Research Center for Nanotechnology and Advanced Materials, Pontificia Universidad Católica de Chile, Av. Vicuña Mackenna 4860, Macul, Santiago 7820436, Chile; 3Energy Center, Pontificia Universidad Católica de Chile, Av. Vicuña Mackenna 4860, Macul, Santiago 7820436, Chile; 4Faculty of Economics and Business, Universitat Oberta de Catalunya, Rambla del Poblenou 156, 08018 Barcelona, Spain; 5School of Environmental Sciences and Sustainability, Faculty of Life Sciences, Universidad Andres Bello, República 440, Santiago 8370076, Chile

**Keywords:** *Aspergillus flavus*, bioremediation, diesel contamination, HDPE, fungal colonization, weathering, halo formation, temperature effect

## Abstract

Plastic containers used for diesel storage represent an underexplored but significant environmental challenge due to hydrocarbon retention and prolonged weathering. This study evaluates the capacity of *Aspergillus flavus* to colonize and grow on high-density polyethylene (HDPE) surfaces contaminated with weathered and fresh diesel residues. Circular plastic samples from HDPE tanks exposed to environmental conditions for over two years (weathered) and for less than two months (non-weathered) were inoculated with *A. flavus* and incubated at 20 °C, 25 °C, and 30 °C. Growth kinetics were assessed through radial expansion and halo formation, quantified via digital imaging and ImageJ analysis. Results showed the most robust fungal growth occurred on weathered diesel-contaminated gray plastics at 30 °C, with colony areas exceeding 350 mm^2^ and halos over 3000 mm^2^. Conversely, white HDPE with fresh diesel showed limited and inconsistent growth, likely due to the presence of volatile hydrocarbons and polymer additives. These findings underscore the critical role of diesel aging and polymer characteristics in shaping fungal adaptability, providing a foundation for the development of environmentally sustainable bioremediation strategies targeting diesel-contaminated HDPE plastics.

## 1. Introduction

Plastic waste generated from the use of tanks for the storage of fuels, lubricants, additives, and other petroleum-derived products represents an increasingly pressing yet underexplored environmental challenge. While specific data on such waste streams are often not disaggregated, it is estimated that a significant proportion of the more than 460 million metric tons of plastics used annually is associated with the energy and transportation sectors, including high-density polyethylene (HDPE) containers for petroleum derivatives. This figure is projected to exceed 1231 million metric tons by 2060 [1].

Among these applications, HDPE tanks used for diesel storage are notable for their chemical resistance, durability, low weight, and ease of manufacturing. They are commonly employed in agriculture, mining, and transportation, with capacities ranging from 200 to over 5000 L. In Chile, these tanks are regulated under Decree No. 160 [2] issued by the Superintendence of Electricity and Fuels. However, after up to two decades of contact with diesel fuel, their recycling becomes highly complex due to the adsorption and diffusion of hydrocarbons into the polymer matrix, compounded by long-term environmental weathering [3].

Conventional cleaning methods for these tanks typically involve washing with water and detergents, which generates hydrocarbon-laden wastewater requiring subsequent treatment. This merely transfers the contamination from the plastic to the effluent, posing additional environmental risks.

Diesel fuel is a complex mixture comprising hundreds to thousands of compounds, including aromatic hydrocarbons (15–40%), linear alkanes (25–50%), and naphthenes (20–40%), with carbon numbers between C9 and C22 [4]. Among these, aromatic compounds such as toluene are particularly persistent and problematic, and are considered representative of this fraction [5]. These substances are chemically stable and tend to accumulate in plastic matrices.

Weathering, driven by prolonged exposure to environmental factors such as UV radiation, atmospheric oxygen, temperature fluctuations, and humidity, induces substantial chemical transformations in diesel residues adhered to HDPE surfaces [6,7]. These changes include progressive increases in viscosity, reductions in aqueous solubility, and declines in bioavailability [8]. For example, photochemical weathering of petroleum at 5 °C has been reported to increase its viscosity more than sixteenfold and decrease its water-soluble fraction by a factor of seven compared to petroleum at 30 °C, with direct implications for its dispersion and treatability [3].

Under the temperate, semi-arid climate of Santiago, Chile—characterized by high seasonal solar radiation—significant changes in the chemical structure of diesel residues on exposed HDPE tanks begin to manifest within 6 to 12 months. During this period, lighter aliphatic hydrocarbons (C10–C14) tend to evaporate, while heavier components (C15–C22) and aromatic compounds undergo partial oxidation, yielding intermediate products such as carboxylic acids and alcohols. Between 12 and 24 months, the chemical profile of the adhered contaminants diverges markedly from that of fresh diesel, becoming dominated by recalcitrant residues, oxidized polar fragments, and low-volatility compounds, thereby altering their environmental behavior and microbial interactions [9].

Moreover, the simultaneous exposure of HDPE to diesel and sunlight may trigger photoxidative processes within the polymer itself, modifying its surface topography, oxidation state, and affinity for microbial colonization and enzymatic action [10]. This combined effect has direct implications for fungal adhesion and metabolism in bioremediation settings involving *Aspergillus flavus*.

Diesel-contaminated plastic waste also poses environmental and human health risks due to emissions of volatile organic compounds and the potential for leaching into soils and aquatic systems. Bioremediation emerges as a viable, cost-effective, and environmentally sustainable alternative to traditional physico-chemical treatment methods [11].

In particular, filamentous fungi such as white-rot fungi (WRF) are noted for their ability to penetrate solid matrices, secrete ligninolytic enzymes, and degrade petroleum hydrocarbons [10]. The concept of fungal-based biocleaning involves using fungal strains to degrade organic contaminants directly on solid surfaces, with minimal water use. This method reduces the generation of contaminated effluents and water consumption, although it presents technological challenges such as maintaining suitable environmental conditions, preserving surface moisture, and selecting strains capable of effective colonization on weathered and contaminated substrates.

Fungal genera such as *Aspergillus* and *Penicillium*—both from the Aspergillaceae family—are recognized for their resilience to extreme environmental conditions and their versatility in industrial and bioremediation applications. However, their application in closed-system biocleaning remains unreported [12,13].

Fungal biodegradation of high-density polyethylene (HDPE) has garnered increasing attention due to the pressing need for sustainable solutions to plastic pollution. Various fungal taxa have demonstrated notable potential in degrading HDPE under laboratory conditions. Among them, *Aspergillus* species, such as *A. niger*, *A. flavus*, *A. terreus*, and *A. fumigatus*, have been widely reported to colonize polyethylene surfaces and induce chemical and morphological changes consistent with biodegradation, including weight loss and oxidation of polymer chains [14,15]. *Penicillium* species, notably *P. citrinum*, *P. chrysogenum*, and *P. oxalicum*, have also been identified as effective agents in HDPE degradation, capable of reducing polymer mass and inducing visible cracks and microstructural damage on plastic films [16]. Similarly, members of the genus *Fusarium*, such as *F. oxysporum* and *F. solani*, have exhibited enzymatic capabilities to attack polyethylene matrices, resulting in measurable mass loss and alterations in infrared spectra. Recent investigations have even isolated effective HDPE-degrading fungi from unique ecosystems, such as mangrove soils (*Aspergillus niger*, *Penicillium citrinum*) and insect gut microbiota (e.g., *Cladosporium halotolerans* from *Galleria mellonella* larvae), underscoring the adaptability and metabolic diversity of fungi in plastic degradation [14,17]. These fungi secrete a wide range of oxidative enzymes, including laccases, peroxidases, and cutinases, which facilitate the breakdown of recalcitrant polymers into metabolizable oligomers. Collectively, these findings highlight the potential of *Aspergillus*, *Penicillium*, *Fusarium*, and related genera in advancing fungal-based strategies for mitigating HDPE pollution.

In the context of fungal biodegradation studies, radial growth rate has been widely used as a kinetic parameter to assess colonization capacity on various substrates [18,19]. Additionally, the formation of halos surrounding the fungal colonies has been recognized as a complementary indicator of metabolic activity, likely associated with the diffusion of extracellular enzymes or secondary metabolites, thus offering insights into their biochemical potential for surface degradation [18].

Since petroleum derivatives such as diesel undergo physicochemical transformations during weathering that alter their toxicity and bioavailability, the development of fungal-based biocleaning technologies must begin by evaluating microbial strains capable of colonizing and growing on contaminated plastics under these conditions. Accordingly, this study aims to assess the growth capacity of the filamentous fungus *Aspergillus flavus* on HDPE surfaces retrieved from discarded diesel storage tanks contaminated with diesel fuel weathered for more than two years. The study also includes a comparative evaluation on HDPE surfaces contaminated with fresh diesel (less than two months of exposure) to identify the environmental and experimental conditions that enhance fungal adaptation and inform potential biocleaning strategies.

## 2. Materials and Methods

### 2.1. Study Objective and Experimental Design

This study aimed to evaluate the ability of the filamentous fungus *Aspergillus flavus* to colonize and grow on high-density polyethylene (HDPE) surfaces contaminated with diesel fuel. Three contamination conditions were considered: HDPE from a tank exposed to fresh diesel for less than two months, HDPE from a tank exposed to weathered diesel residues for over two years, and HDPE from a new, unused tank serving as an uncontaminated control. For each condition, six HDPE sections were collected and fabricated into 18 circular plastic discs, allowing for triplicate assays under each experimental setting. The samples were incubated at three distinct temperatures (20 °C, 25 °C, and 30 °C) under controlled environmental conditions, with relative humidity maintained at approximately 90%.

Fungal growth was monitored to assess the influence of temperature on colonization. Radial growth rate (RGR) was used as a kinetic metric to evaluate fungal expansion across HDPE surfaces, following established methodologies [18].

### 2.2. Chemical Characterization of Diesel Fuel in Study Samples

#### 2.2.1. Virgin Diesel

Samples of virgin diesel were collected directly from service stations in the Metropolitan Region of Santiago, Chile. Sampling was performed using sterilized and labeled 1 L amber glass containers, which were stored at 4 °C until analysis. The samples were diluted 1:10 with HPLC-grade n-hexane and analyzed via gas chromatography-mass spectrometry (GC-MS) using an Agilent 7890B GC system coupled with a 5977B MSD detector and an HP-5MS UI column (30 m × 0.25 mm × 0.25 µm). The oven temperature gradient was set between 40 °C and 300 °C. Compound identification was conducted with reference to the NIST library, focusing on the relative quantification of n-alkanes, aromatic hydrocarbons, naphthenes, and heavy fractions.

#### 2.2.2. Weathered Diesel

For the analysis of weathered diesel, HDPE tanks exposed to the environment for more than two years were selected from industrial and agricultural disposal sites in the Metropolitan Region. Contaminated plastic fragments (approximately 5 g) were collected from the internal walls of the tanks and stored in sealed bags at 4 °C. Residue extraction was performed using solid-phase microextraction (SPME) with DVB/CAR/PDMS fibers (Supelco, Bellefonte, PA, USA). Samples were heated to 60 °C for 30 min in sealed vials, and thermal desorption was conducted in the injector of the same GC-MS system (Agilent 7890B GC coupled with a 5977B MSD detector and an HP-5MS UI column; Agilent Technologies, Santa Clara, CA, USA).

### 2.3. Preparation of HDPE Samples

Two types of contaminated plastic matrices were selected for experimental evaluation. The first group consisted of six samples extracted from the internal side walls of three gray HDPE tanks, each with a capacity of 1000 L. These tanks had been in service for seven years and had been exposed to the environment for more than two years without any subsequent maintenance or intervention. The second group comprised six samples from three white HDPE tanks of the same brand, capacity, and service duration but with significantly less environmental exposure, under two months, referred to as non-weathered.

Both tank types exhibited visible diesel fuel residues, remained intact (without perforations), and had functional closure systems. All were retrieved from authorized industrial disposal sites in the Metropolitan Region of Santiago, Chile. Sampling focused on the lower third of the internal side walls, where higher accumulation and adherence of hydrocarbon contaminants were observed.

Additionally, six control samples were obtained from a new, uncontaminated gray HDPE diesel fuel tank of identical characteristics. All samples were processed in triplicate. In total, 18 sets of three circular plastic discs, each 65 mm in diameter, were prepared using a carbon steel hole saw attached to an industrial drill, ensuring uniformity in dimensions and exposed area.

Prior to inoculation with *Aspergillus flavus*, all samples underwent a microbiological decontamination protocol using ultraviolet (UV) radiation, with 48 h exposure on each side. The effectiveness of UV-C decontamination (254 nm, 30 min per side) was verified by incubating a subset of sterilized HDPE discs on PDA without inoculation for 7 days at 25 °C. No microbial growth was observed, confirming the efficacy of the decontamination procedure. This procedure, based on previous studies [16,19,20] aimed to eliminate pre-existing microorganisms and ensure that observed growth was exclusively attributable to the inoculated strain.

### 2.4. Preparation of Culture Medium

For the cultivation of *Aspergillus flavus*, potato dextrose agar (PDA) medium was used, prepared from a standardized commercial formulation (Difco™ PDA, BD Diagnostics, NJ, USA, Cat. No. 213400). A total of 39 g of dehydrated powder was dissolved in 1 L of distilled water and sterilized in an autoclave at 121 °C for 20 min. Once cooled to 50 °C, the medium was applied with a sterile brush onto the surface of the plastic discs, ensuring a thin and homogeneous coating. The plastic discs were then placed in sterile 10 cm diameter Petri dishes for incubation.

### 2.5. Inoculation and Incubation

The strain used in this study was *Aspergillus flavus*, selected for its recognized capacity to metabolize hydrocarbons and its tolerance to adverse environmental conditions [21]. This strain was obtained from the fungal bank of the Renewable Energies and Waste Laboratory (LERR-UC) at the Pontificia Universidad Católica de Chile and maintained under refrigeration until experimental use. The genetic sequence of the strain employed in this study has been published in [22].

The selection of incubation temperatures (20 °C, 25 °C, and 30 °C) was based on the known thermal tolerance of *Aspergillus flavus*, which ranges approximately between 12 °C and 48 °C, with a general physiological optimum near 37 °C under laboratory conditions. Additionally, the production of secondary metabolites such as aflatoxins is higher between 25 °C and 33 °C, with aflatoxin B1 (AFB1) peaking at 33 °C and AFB2 at temperatures around 25–30 °C, provided favorable water activity conditions exist [23]. The presence of aflatoxins was not evaluated in this study.

However, this study prioritized an environmental applicability approach. Considering that fungal-based biocleaning technologies could be implemented in industrial, agricultural, or exposed disposal environments, it was deemed essential to evaluate the efficacy of the fungus at temperatures compatible with ambient conditions, without the need for artificial thermal control. In this context, temperatures of 20 °C and 25 °C were included to represent real-world usage scenarios, allowing exploration of the potential of *A. flavus* in sustainable and low-energy decontamination systems, even operating below its optimal toxin production range.

Inoculation involved depositing a portion of active *A. flavus* mycelium at the center of each plastic sample previously coated with PDA medium. The samples were placed in sterile glass Petri dishes and incubated in a forced convection incubator (Memmert IF55, Memmert GmbH, Schwabach, Germany), programmed at three experimental temperatures: 20 ± 1 °C, 25 ± 1 °C, and 30 ± 1 °C. These conditions were selected based on the literature highlighting their influence on enzymatic activity and mycelial expansion in fungal bioremediation processes [8,10]. The incubation period extended for 15 days, maintaining constant relative humidity and internal ventilation within the equipment.

### 2.6. Monitoring and Evaluation of Fungal Growth

Digital images were captured every three days to monitor the progression of fungal colonization. The method described by Khan [18] was employed to measure the radial diameter of mycelial growth, thereby estimating the growth rate as a kinetic parameter of the biocleaning process. Images were analyzed using ImageJ v1.54 software, calibrated with a physical scale printed on each Petri dish. The radial diameter of the mycelium was measured along perpendicular axes, obtaining an average per sample.

## 3. Results

### 3.1. Composition of Commercial Diesel Fuel

Gas chromatographic analysis of commercial diesel fuel collected from service stations in Santiago revealed a characteristic chemical profile dominated by mid- and long-chain n-alkanes, aromatic hydrocarbons, and naphthenes. The most abundant components were n-alkanes ranging from C15 to C25, collectively accounting for over 45% of the total mass. Mono- and polycyclic aromatic hydrocarbons (including BTEX and trace levels of PAHs) were also detected, representing approximately 20% of the composition. An additional 13% of the sample mass consisted of unidentified residues, including heavier compounds and fuel additives. These data are summarized in Table 1.

### 3.2. Composition of Weathered Diesel Fuel Extracted from HDPE Fuel Container Surfaces

The diesel fuel extracted from HDPE plastic fragments exhibited a significantly altered composition due to prolonged weathering. A marked reduction in volatile compounds (C10–C14) and light aromatic hydrocarbons was observed, while the proportion of heavier n-alkanes (C21–C25) and unidentified residuals increased substantially, accounting for over 80% of the sample. These results indicate extensive loss of components through evaporation, oxidation, and both chemical and biological transformation over time. Detailed results are presented in Table 1 and compared to commercial diesel fuel composition.

### 3.3. Fungal Growth Dynamics on Plastic Substrates

The analysis of *Aspergillus flavus* growth revealed effective colonization across all evaluated HDPE plastic substrates, both in the presence and absence of weathered diesel fuel residues. Throughout the incubation period, a steady increase in the colonized area by fungal hyphae was observed, along with peripheral halo formation, reaching over 300 mm^2^ in radial expansion and up to 3000 mm^2^ in halo extension, depending on the experimental conditions. These results suggest a suitable physiological adaptation of the fungus to the tested substrates.

#### 3.3.1. Comparative Fungal Behavior at 30 °C According to Plastic Substrate Type

At 30 °C, marked differences in *Aspergillus flavus* behavior were observed depending on the type of plastic matrix and its contamination status. The weathered and diesel-contaminated gray HDPE, as shown in Table 2, proved to be the most favorable substrate, reaching the highest values of mycelial expansion (colony area averaging up to 355.8 mm^2^ ± 125.13) and halo formation (up to 3109.66 mm^2^). This suggests a positive synergy between contaminant weathering and thermal conditions.

In contrast, the non-weathered, diesel-contaminated white HDPE, shown in Table 2, exhibited the most limited fungal growth, with average colony areas of just 20.1 mm^2^ ± 14.16 and inconsistent or minimal halo formation. These results indicate a likely combination of toxicity from commercial diesel and a possible inhibitory effect of additives in the white HDPE matrix.

The control group, represented by uncontaminated gray HDPE (Table 2), displayed an intermediate growth pattern, with more stable mycelial expansion and moderate colony areas (average of 103.15 mm^2^ ± 41.69). Interestingly, halo formation in the control samples reached values comparable to those observed on weathered diesel-contaminated plastic, suggesting strong metabolic activity even in the absence of hydrocarbons, likely supported by the nutritional contribution of the PDA medium.

#### 3.3.2. Comparison of Fungal Behavior at 25 °C by Plastic Matrix Type

At 25 °C, the behavior of *Aspergillus flavus* followed the same trend observed at 30 °C, although with a general reduction in colony growth and halo formation areas. The gray HDPE plastic contaminated and weathered for more than two years, as shown in Table 3, remained the most favorable substrate, with mean colony areas reaching up to 74.19 mm^2^ ± 25.08 and halo extensions up to 2623.59 mm^2^. These results confirm that diesel weathering reduces its toxicity and enhances the fungus’s metabolic activity, even at temperatures close to ambient.

The white HDPE plastic contaminated with non-weathered diesel (less than two months of exposure), as presented in Table 3, again showed limited fungal growth, with mean colony areas ranging from 23.79 mm^2^ to 38.17 mm^2^, and halo formation being sporadic and highly variable (some exceeding 1000 mm^2^, others completely absent). This heterogeneity suggests that the presence of fresh hydrocarbons and possibly plastic additives in the white HDPE continued to negatively impact fungal colonization.

The control samples (uncontaminated gray HDPE), shown in Table 3, exhibited moderate and consistent growth. Colonies reached mean areas of up to 55.14 mm^2^ ± 13.24, with halos exceeding 1400 mm^2^ in some replicates. While these values were lower than those observed at 30 °C, they indicate that the substrate supports active fungal metabolism even in the absence of hydrocarbons, confirming its suitability as an experimental reference baseline.

#### 3.3.3. Comparison of Fungal Behavior at 20 °C According to Plastic Matrix Type

At 20 °C, the behavior of *Aspergillus flavus* followed the same pattern observed at higher temperatures, although with an overall reduction in mycelial growth rate and halo formation. The weathered and diesel-contaminated gray plastic (see Table 4) remained the most favorable matrix, reaching an average colony area of 78.39 mm^2^ ± 23.36 and halo areas up to 1640.63 mm^2^ ± 1044.26. Although these figures were lower than those observed at 25 °C and 30 °C, they confirm that the weathered contaminant continues to offer conditions compatible with fungal activity, even under less favorable thermal conditions.

The unweathered, diesel-contaminated white plastic (see Table 4) once again showed the most limited growth. Average colony areas ranged from 19.24 mm^2^ to 33.85 mm^2^, with halos absent in several replicates or very small, with maximum values around 871.23 mm^2^. The data dispersion and limited halo formation reinforce the hypothesis that both the composition of commercial diesel and the additives present in white HDPE negatively impact *A. flavus* colonization capacity.

In the case of the control (uncontaminated gray plastic, see Table 4), fungal growth was moderate and relatively homogeneous, with average colony areas of 42.30 mm^2^ ± 24.32 and halo areas reaching up to 1538.96 mm^2^ ± 949.21. Although these values were lower than those recorded at higher temperatures, they indicate that the fungus maintains a stable growth capacity, confirming the utility of this matrix as a comparative reference under varying thermal conditions.

#### 3.3.4. Analysis of Experimental Variability Across Treatments

To complement the evaluation of *Aspergillus flavus* behavior on the different plastic matrices, coefficients of variation (CV) were calculated for colony area in each experimental combination (matrix × temperature). This allowed for quantification of relative data dispersion and the identification of patterns of homogeneity or heterogeneity among replicates.

At 30 °C, the treatment involving weathered and diesel-contaminated gray plastic exhibited a CV of 35.2%, indicating moderate variability among replicates, likely attributable to microenvironmental differences or heterogeneity in the aged plastic substrate. In contrast, the unweathered diesel-contaminated white plastic presented a high CV of 58.3%, reflecting considerable data dispersion and a potential sensitivity of the fungus to this specific matrix. The control treatment—uncontaminated gray plastic—showed an intermediate CV of 40.4%.

At 25 °C, a similar trend was observed: weathered diesel-contaminated gray plastic maintained moderate variability (CV = 33.8%), while the white unweathered plastic again demonstrated high variability (CV = 55.1%), reinforcing its erratic behavior. The control group displayed the lowest CV in this series (24.0%), indicating greater consistency among replicates.

At 20 °C, the pattern was again consistent: the CV was 29.8% for weathered diesel-contaminated gray plastic, 50.2% for unweathered diesel-contaminated white plastic, and 34.2% for the control. These values suggest that even under less favorable thermal conditions, the weathered and contaminated matrix promotes more predictable fungal growth compared to commercial diesel-contaminated white plastic.

These findings indicate that experimental variability is influenced not only by incubation temperature but also by the matrix type and the degree of contaminant weathering. The most heterogeneous conditions were consistently associated with the white plastic containing commercial diesel, whereas weathered or uncontaminated substrates offered more stable environments for *A. flavus* growth.

The coefficient of variation (CV) is a statistical metric that expresses the relative dispersion of a dataset with respect to its mean, enabling the comparison of variability across experimental conditions with differing absolute magnitudes. It is calculated as the ratio of the standard deviation to the mean, expressed as a percentage. In microbiological studies, CV values below 30% are typically interpreted as indicative of high experimental homogeneity, whereas values between 30% and 50% represent acceptable variability. CVs exceeding 50% reflect substantial data dispersion, often associated with complex interactions or non-uniform responses by the organism under study.

In the present work, the highest CV values were systematically observed in treatments involving non-weathered, diesel-contaminated white plastic, consistently exceeding 50% at all tested temperatures. This finding suggests a combination of physiological sensitivity of the fungus to commercial diesel and potential interference from physicochemical characteristics of the white polymer, such as the presence of antioxidant additives or pigments, which may hinder mycelial colonization in a heterogeneous manner across replicates.

These results reinforce the importance of considering both the composition of the contaminant and the plastic substrate in the design of fungus-based biocleaning strategies.

#### 3.3.5. Influence of Incubation Temperature on Fungal Colonization

Incubation temperature had a decisive impact on the behavior of *Aspergillus flavus*, affecting both the mycelial growth area and the formation of visible extracellular metabolites (halos). In general, a trend of increasing mycelial development was observed as incubation temperature rose from 20 °C to 30 °C, particularly on substrates containing weathered diesel.

In the gray plastic matrices contaminated with weathered diesel, increasing temperature led to progressive increases in the average colony area—from 78.39 mm^2^ at 20 °C, to 74.19 mm^2^ at 25 °C, and up to 355.8 mm^2^ at 30 °C. Similarly, halo expansion followed this trend, reaching a maximum area of 3109.7 mm^2^ at 30 °C. These results suggest a positive synergy between elevated thermal conditions and the presence of partially degraded contaminants, which were likely more bioavailable as carbon sources.

In contrast, white plastic samples contaminated with non-weathered diesel exhibited severely limited growth at all temperatures. This matrix proved the least favorable for *A. flavus*, with consistently low colony areas (typically under 50 mm^2^) and erratic halo formation. The lack of diesel weathering, combined with the potential inhibitory effects of HDPE additives typical in fuel storage tanks, may have created chemically adverse conditions for both mycelial colonization and enzymatic expression.

The control samples (uncontaminated gray plastic) displayed moderate and stable growth. Although the average colony area increased with temperature (from 42.3 mm^2^ to 103.15 mm^2^), halo formation was variable but significant, indicating metabolic activity even in the absence of hydrocarbons. This highlights the capacity of PDA medium to support baseline fungal growth, as well as the influence of thermal gradients on fungal physiology.

In summary, incubation temperature acts as a critical modulating factor, but its effects depend strongly on the substrate type and the weathering status of the contaminant. A temperature of 30 °C was consistently the most favorable for *A. flavus* growth in the presence of weathered diesel, whereas temperature effects were less pronounced or even adverse in matrices containing commercial diesel.

#### 3.3.6. Temporal Evolution of Mycelial Growth and Halo Formation

The temporal analysis of *Aspergillus flavus* growth on different plastic matrices revealed distinct patterns based on substrate and incubation temperature. Overall, a progressive increase in total colony and halo areas was observed over the 17-day incubation period, although with notable differences among treatments.

At 30 °C (see Figure 1), the gray plastic matrix contaminated with weathered diesel demonstrated the most pronounced and sustained progression over time, both in radial mycelial expansion and in the formation of diffusible extracellular metabolites. Colony areas increased in a near-linear fashion from 10 mm^2^ on day 0 to 355.8 mm^2^ by day 17, while halo areas reached up to 2823.4 mm^2^ during the same period. This pattern reflects highly favorable conditions for fungal growth and metabolism, with relatively low measurement errors across time points, suggesting robust reproducibility among replicates. 

At 25 °C (see Figure 2), the same substrate exhibited effective growth, albeit with a slight reduction in expansion rate, particularly during the final days of the experiment. Colony areas reached 190 mm^2^ and halos 1437.97 mm^2^, indicating a deceleration compared to the values observed at 30 °C. This behavior suggests that, although the temperature still supports significant activity, the metabolic performance of the fungus is partially constrained.

In contrast, the white plastic matrices contaminated with non-weathered diesel exhibited the lowest fungal development under all conditions, with slower growth and greater data dispersion, as reflected by the standard errors (Table 5). At 20 °C and 25 °C (see Figure 2 and Figure 3), increases in area were low and erratic, with colony sizes barely exceeding 45 mm^2^ and halos showing high variability and little sustained expression over time. This dynamic supports the hypothesis that both the lack of contaminant weathering and the nature of the white polymer (possibly containing inhibitory additives) negatively impacted the colonization and metabolism of *A. flavus*.

In turn, the uncontaminated gray plastic (control) exhibited progressive and more stable growth, particularly in the formation of halos. At 20 °C and 25 °C (see Figure 2 and Figure 3), these reached total areas of 1987.5 mm^2^ and 1538.96 mm^2^, respectively, while colony areas approached 100 mm^2^, with lower relative variability. This suggests that although the absence of diesel does not stimulate fungal metabolism as a carbon source, the plastic substrate alone does not significantly inhibit its growth.

Overall, these results highlight that the fungal growth of *A. flavus* is influenced not only by the availability of contaminants as a substrate but also by factors such as the prior weathering of diesel, incubation temperature, and the composition of the base polymer. The incubation temperature of 30 °C was clearly the most favorable, particularly when combined with weathered gray plastic matrices, thereby establishing a useful framework for future applied studies on fungal-based biocleaning.

#### 3.3.7. Influence of Temperature and Substrate Type on Fungal Growth: ANOVA Results

At 30 °C, the analysis of variance (ANOVA) revealed statistically significant differences in the colony growth area of *Aspergillus flavus* among the different types of HDPE evaluated (contaminated with commercial diesel, weathered diesel, and clean control). The obtained *p*-value was 0.0018, well below the conventional threshold of 0.05, indicating that the observed differences are unlikely to be due to random variation. Furthermore, the F-statistic of 8.09 supports the presence of a substantial effect of substrate type on fungal development under elevated thermal conditions. These results suggest that, at 30 °C, the nature of the contaminant present on the HDPE surface plays a decisive role in the fungus’s colonization capacity.

As temperature increased from 20 °C to 25 °C and subsequently to 30 °C, the differences in colony growth area of *Aspergillus flavus* among the various HDPE types became progressively more statistically significant. At 20 °C, no significant differences were detected between the substrate conditions. However, at 25 °C, a near-significant trend emerged, and by 30 °C, the differences were clearly significant, indicating a temperature-dependent enhancement of the fungus’s discriminatory growth response to the physicochemical characteristics of the HDPE surfaces.

## 4. Qualitative Analysis of Photographic Images of Fungal Cultures

The qualitative analysis of the images captured throughout the experiment revealed marked morphological differences in the colonization patterns of *Aspergillus flavus*, depending on the plastic matrix and the incubation temperature.

### 4.1. Samples of Gray Plastic Contaminated with Weathered Diesel Fuel

In samples of gray plastic contaminated with weathered diesel fuel, the colonies exhibited a continuous and homogeneous radial expansion pattern across all evaluated temperatures. At 20 °C, colonies displayed well-defined edges and halos of low to moderate peripheral intensity (Figure 4A), consistent with the average halo area of 660.4 mm^2^ recorded under this condition. At 25 °C (Figure 4B), the mycelium appeared denser and the halos expanded significantly, reflecting the increases quantified in the average halo values (Table 5).

At 30 °C (Figure 4C), the highest degree of mycelial expansion was observed. Colonies presented a compact and regular structure, with broad, diffuse, and well-distributed halos, in line with the maximum halo areas exceeding 3000 mm^2^ recorded under this condition (Table 2). These observations support the quantitative results and suggest that the fungus not only tolerates but adapts favorably to the presence of the contaminant in this matrix, particularly under elevated thermal conditions.

### 4.2. White Plastic Contaminated with Non-Weathered Diesel Fuel

In samples of white plastic contaminated with non-weathered diesel fuel, *Aspergillus flavus* exhibited markedly more irregular and fragmented growth compared to other matrices. At 20 °C (Figure 5A), colonies showed diffuse edges and discontinuous expansion, consistent with the low radial growth values observed (average of 47.68 mm^2^, see Table 4 and with the absence or minimal development of peripheral halos.

At 25 °C (Figure 5B), a slight increase in mycelial density was observed, although irregularity at the colony margins persisted, in line with the high variability recorded in growth areas. At 30 °C (Figure 5C), the pattern was even more restricted: colonies were small, with poorly defined edges and low cohesiveness, while halos, when present, were limited in size, with averages below 300 mm^2^ (Table 3).

These visual observations reinforce the hypothesis that this matrix offers less favorable conditions for fungal development, even in the presence of the contaminant. This may be attributed to specific physicochemical characteristics of the white plastic that hinder adhesion, metabolite diffusion, or access to hydrocarbons, in addition to potential inhibitory effects of the fresh, non-weathered diesel fuel stored in the tank.

### 4.3. Gray Plastic Without Contamination Used as Control

In the control samples, consisting of new, uncontaminated gray plastic, *Aspergillus flavus* exhibited a more orderly and symmetrical growth pattern compared to contaminated white plastic, although the overall expansion was less than on contaminated gray plastic. At 20 °C (Figure 6A), colonies showed uniform radial growth with well-defined edges, while halos were less intense and smaller in size than those observed on contaminated gray plastic.

At 25 °C (Figure 6B), a slight increase in mycelial density and halo extension was recorded, consistent with the growth increase observed in the quantitative halo area data. At 30 °C (Figure 6C), some replicates showed broader expansion and more compact structures, although not all reached the level of dispersion and coverage observed in contaminated samples, indicating some metabolic limitation in the absence of weathered hydrocarbon.

These observations confirm that while PDA medium supports the growth of *A. flavus* on uncontaminated plastic surfaces, the presence of contaminants from weathered diesel fuel on HDPE tank surfaces stimulates more vigorous mycelial development, particularly on compatible matrices like gray plastic.

## 5. Conclusions

The results of this study confirm the potential of *Aspergillus flavus* as an efficient fungal agent for colonizing and growing on high-density polyethylene (HDPE) surfaces contaminated with diesel fuel. The most notable mycelial development was observed on gray plastic samples with diesel residues that had undergone natural weathering for more than two years under temperate environmental conditions, such as those found in Santiago, Chile. Incubation at 30 °C resulted in a significant increase in both biomass and the formation of diffusible metabolites, suggesting a synergistic interaction between contaminant aging and incubation temperature. These findings are consistent with the observations of Bai [10], who highlight the thermal sensitivity of fungal metabolism in bioremediation contexts.

Moreover, the presence of weathered diesel-derived contaminants did not inhibit *A. flavus* growth; rather, it appeared to stimulate metabolic activity, likely due to the increased availability of higher molecular weight oxidized compounds, which are more readily metabolized by the fungus’s enzymatic machinery. This behavior aligns with previous reports describing *A. flavus* as a microorganism with a high capacity for adaptation and degradation of recalcitrant organic pollutants [19,24].

The methodology employed, based on kinetic analysis of radial growth and quantification of halo formation, proved robust for assessing the dynamics of fungal colonization. As previously noted by Park [24], these metrics are useful for characterizing fungal metabolic activity and its responses to various environmental conditions or contaminated substrates, providing a replicable framework for future studies.

A noteworthy observation from this study was the limited colonization of white plastic samples recently contaminated with diesel (<2 months of exposure). This behavior may be explained by the higher proportion of volatile and toxic compounds, such as BTEX, present in commercial diesel, along with the higher concentration of plastic additives (e.g., antioxidants and inorganic pigments) that have not yet degraded through weathering. These factors appear to exert a stronger inhibitory effect than the contaminant itself, suggesting that the efficiency of bioremediation depends not only on the fungus and the hydrocarbon but also on the properties of the polymer substrate and its degree of environmental aging.

Overall, this study provides robust experimental evidence supporting the feasibility of using *Aspergillus flavus* in fungal-based cleaning strategies for diesel-contaminated plastic waste. The findings lay the groundwork for developing sustainable and scalable technologies that address the urgent need for eco-friendly and cost-effective solutions in managing waste from industrial and agricultural activities.

Despite the controlled nature of this study, certain limitations should be considered when interpreting the results. The sample size, while consistent with preliminary biodegradation studies, may limit the statistical resolution of comparisons across treatments. Although temperature and humidity were regulated using an integrated control system within the incubator, minor fluctuations may still occur and should be acknowledged. Moreover, while the use of HDPE samples contaminated under real-world weathering conditions adds ecological realism, extrapolating laboratory outcomes to field environments remains challenging due to the complexity of biotic and abiotic interactions. Future research should explore these dynamics in situ or under semi-controlled outdoor conditions.

## Figures and Tables

**Figure 1 microorganisms-13-01418-f001:**
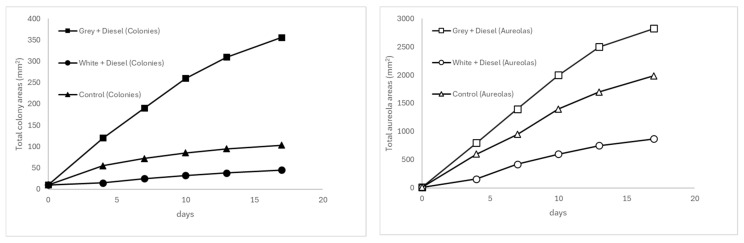
Temporal evolution of fungal colony (**left**) and halo areas (**right**) at 30 °C.

**Figure 2 microorganisms-13-01418-f002:**
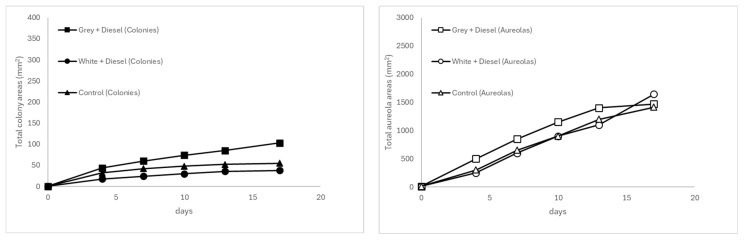
Temporal evolution of fungal colony (**left**) and halo areas (**right**) at 25 °C.

**Figure 3 microorganisms-13-01418-f003:**
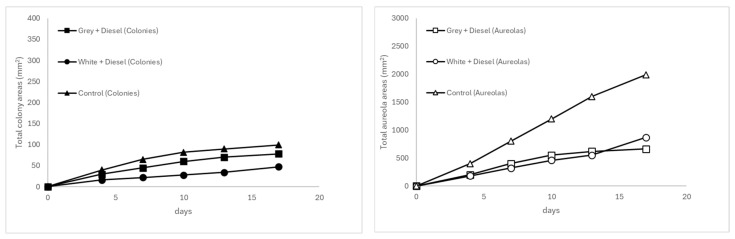
Temporal evolution of fungal colony (**left**) and halo areas (**right**) at 20 °C.

**Figure 4 microorganisms-13-01418-f004:**
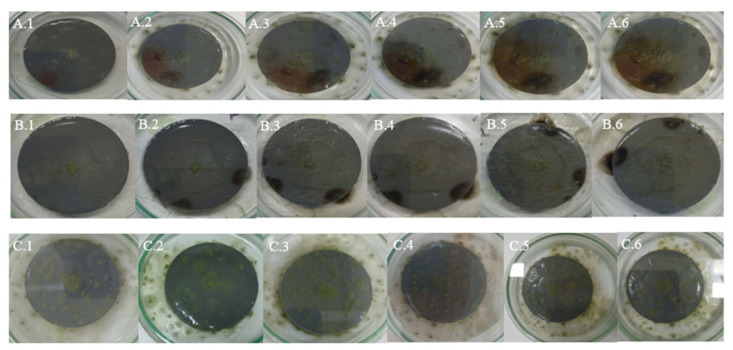
Growth of *Aspergillus flavus* on gray plastic samples contaminated with weathered diesel fuel at 20 °C (**A**), 25 °C (**B**), and 30 °C (**C**).

**Figure 5 microorganisms-13-01418-f005:**
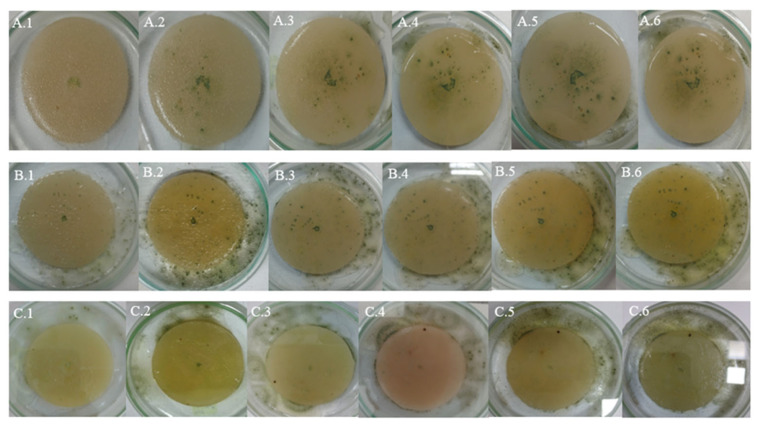
Growth of *Aspergillus flavus* on white plastic samples contaminated with non-weathered diesel fuel at 20 °C (**A**), 25 °C (**B**), and 30 °C (**C**).

**Figure 6 microorganisms-13-01418-f006:**
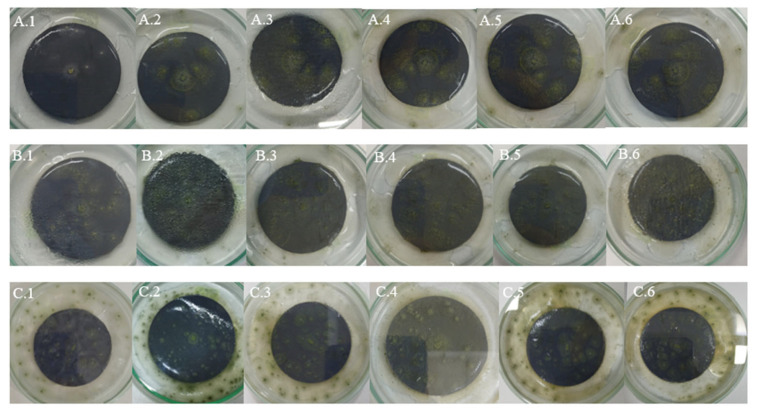
Growth of *Aspergillus flavus* on uncontaminated gray plastic samples at 20 °C (**A**), 25 °C (**B**), and 30 °C (**C**).

**Table 1 microorganisms-13-01418-t001:** Estimated chemical composition of commercial diesel fuel samples and weathered diesel fuel extracted from HDPE plastic containers based on GC-MS analysis.

	Commercial Diesel		Weathered Diesel	
Component	% w/w	Error (%)	% w/w	Error (%)
n-Alkanes C10–C14	8.5	0.38	0.5	0.1
n-Alkanes C15–C20	20.0	0.42	5.0	0.3
n-Alkanes C21–C25	25.0	0.39	8.0	0.4
Aromatic hydrocarbons (BTEX)	15.0	0.16	0.2	0.05
Naphthenes	12.0	0.31	3.0	0.2
Polycyclic aromatic hydrocarbons (PAHs)	5.0	0.32	1.5	0.1
Pristane and phytane	1.5	0.19	0.7	0.05
Unidentified fraction (residuals)	13.0	0.24	81.1	0.5

**Table 2 microorganisms-13-01418-t002:** Total colony and halo areas on samples of weathered HDPE gray plastic contaminated with diesel fuel (weathered samples), unweathered white HDPE plastic contaminated with diesel fuel (unweathered samples) and unweathered and uncontaminated gray HDPE plastic samples (control samples), incubated at 30 °C.

	Weathered Samples		Unweathered Samples		Control Samples	
Sample (30 °C)	Mean ± SD (mm^2^)	Mean ± SD (mm^2^)	Mean ± SD (mm^2^)	Mean ± SD (mm^2^)	Mean ± SD (mm^2^)	Mean ± SD (mm^2^)
1	168.01 ± 22.3	1647.17 ± 463.91	11.72 ± 4.12	n.a.	101.07 ± 56.03	n.a.
2	154.62 ± 63.01	1632.86 ± 470.5	18.54 ± 9.16	257.85 ± 311.29	108.89 ± 61.16	1788.58 ± 178.13
3	275.51 ± 30.16	2363.83 ± 488.25	18.31 ± 9.55	273.85 ± 296.04	101.12 ± 52.90	1951.67 ± 426.44
4	269.97 ± 14.44	2347.39 ± 580.89	18.31 ± 9.55	273.85 ± 296.04	103.15 ± 41.69	2068.26 ± 320.96
5	355.8 ± 125.13	2823.45 ± 365.65	18.23 ± 15.3	786.62 ± 597.94	103.15 ± 41.69	2068.26 ± 320.96
6	355.8 ± 125.13	2823.45 ± 365.65	20.1 ± 14.16	1036.23 ± 454.27	103.15 ± 41.69	2290.89 ± 698.34
	Colony area	Halo area	Colony area	Halo area	Colony area	Halo area

**Table 3 microorganisms-13-01418-t003:** Total colony and halo areas on samples of weathered HDPE gray plastic contaminated with diesel fuel (weathered samples), unweathered white HDPE plastic contaminated with diesel fuel (unweathered samples) and unweathered and uncontaminated gray HDPE plastic samples (control samples), incubated at 25 °C.

	Weathered Samples		Unweathered Samples		Control Samples	
Sample (25 °C)	Mean ± SD (mm^2^)	Mean ± SD (mm^2^)	Mean ± SD (mm^2^)	Mean ± SD (mm^2^)	Mean ± SD (mm^2^)	Mean ± SD (mm^2^)
1	43.88 ± 6.87	754.65 ± 188.26	25.63 ± 5.04	n.a.	42.3 ± 24.32	n.a.
2	64.01 ± 15.76	1640.63 ± 1044.26	23.79 ± 7.49	781.98 ± 214.36	42.3 ± 24.32	587.02 ± 313.9
3	47.88 ± 8.18	1344.78 ± 723.93	35.55 ± 19.23	598.39 ± 352.26	54.21 ± 32.73	1538.96 ± 949.21
4	47.88 ± 8.18	1344.78 ± 723.93	34.49 ± 17.11	1039.12 ± 728.00	53.86 ± 31.24	1628.69 ± 899.63
5	72.19 ± 27.19	1437.97 ± 260.99	38.17 ± 17.47	814.66 ± 737.94	55.14 ± 13.24	1268.75 ± 126.77
6	74.19 ± 25.08	1437.97 ± 260.99	45.24 ± 26.37	870.22 ± 984.13	49.15 ± 23.26	1229.19 ± 174.54
	Colony area	Halo area	Colony area	Halo area	Colony area	Halo area

**Table 4 microorganisms-13-01418-t004:** Total colony and halo areas on samples of weathered HDPE gray plastic contaminated with diesel fuel (weathered samples), unweathered white HDPE plastic contaminated with diesel fuel (unweathered samples) and unweathered and uncontaminated gray HDPE plastic samples (control samples), incubated at 20 °C.

	Weathered Samples		Unweathered Samples		Control Samples	
Sample (20 °C)	Mean ± SD (mm^2^)	Mean ± SD (mm^2^)	Mean ± SD (mm^2^)	Mean ± SD (mm^2^)	Mean ± SD (mm^2^)	Mean ± SD (mm^2^)
1	29.31 ± 7.82	314.65 ± 173.9	14.78 ± 10.97	n.a.	24.24 ± 10.51	n.a.
2	119.42 ± 56.37	735.88 ± 428.1	33.92 ± 20.04	157.17 ± 133.89	69.55 ± 26.68	1024.85 ± 86.90
3	90.86 ± 52.75	625.89 ± 343.95	53.27 ± 30.78	n.a.	69.55 ± 26.68	1024.85 ± 86.90
4	78.39 ± 23.36	660.4 ± 152.58	54.29 ± 29.05	n.a.	128.11 ± 100.07	1568.5 ± 570.31
5	78.39 ± 23.36	660.4 ± 152.58	46.59 ± 24.28	498.91 ± 174.39	95.48 ± 34.63	1896.73 ± 248.84
6	78.39 ± 23.36	660.4 ± 152.58	47.68 ± 22.43	526.83 ± 134.9	99.47 ± 33.99	1987.47 ± 174.92
	Colony area	Halo area	Colony area	Halo area	Colony area	Halo area

**Table 5 microorganisms-13-01418-t005:** Coefficients of variation (CV%) by plastic matrix and temperature.

Temperature (°C)	Matrix Type	CV (%)
20	Gray + Diesel	29.8
20	White + Diesel	50.2
20	Control (Gray)	34.2
25	Gray + Diesel	33.8
25	White + Diesel	55.1
25	Control (Gray)	24.0
30	Gray + Diesel	35.2
30	White + Diesel	58.3
30	Control (Gray)	40.4

## Data Availability

The data supporting the findings of this study are included within the article.

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
