# Peer review of "Evaluation of Aspergillus flavus Growth on Weathered HDPE Plastics Contaminated with Diesel Fuel"

_microorganisms, 2025, doi:10.3390/microorganisms13061418_

Round 1

Reviewer 1 Report

Comments and Suggestions for Authors

Journal: Microorganisms (ISSN 2076-2607)

Manuscript ID: microorganisms-3659489

Type: Article

Title: Evaluation of Aspergillus flavus Growth on Weathered HDPE Plastics Contaminated with Diesel Fuel

This manuscript presents a timely and well executed experimental study assessing the colonization capacity of Aspergillus flavus on diesel-contaminated HDPE plastics under varying weathering and thermal conditions. The work addresses a notable gap in bioremediation research, particularly in the fungal degradation of aged hydrocarbon residues on polymeric substrates. The structure of the article is coherent, the methodology is robust, and the findings are meaningful for both applied and fundamental environmental microbiology.

Introduction around lines 87-98: The introduction is well written, but it could benefit from a brief review of existing studies on plastic biodegradation by fungi, not just hydrocarbons.

Material and Methods: How were the replicates distributed across the experimental treatments? How was relative humidity controlled and monitored during incubation? Was the effectiveness of the UV decontamination treatment on plastic discs verified? How was the application of PDA medium on the plastic discs standardized? What specific identifier does the Aspergillus flavus strain used in the study have? Aspergillus flavus is known for producing aflatoxins, which are toxic and carcinogenic compounds. It is unclear whether the presence or the risk of aflatoxin production was evaluated in this study.

Results: In the sections comparing fungal behavior on different types of plastic matrices and at various temperatures, mean values, standard deviations, and, in some cases, coefficients of variation are reported. However, it is not mentioned whether the observed differences between groups were subjected to statistical significance testing (e.g., ANOVA or t-test). The graphs showing the temporal evolution of colonies and halos (Figures 2-4) are very crowded, which makes interpretation difficult.

Conclusions: There is no reflection on the experimental limitations, such as sample size, humidity control, or the extrapolation of results to real-world environments.

I await an improved version of this manuscript for review.

Best regards.

Comments on the Quality of English Language

The English could be improved to more clearly express the research.

Author Response

Dear Reviewer 1,

We sincerely thank you for your careful and constructive evaluation of our manuscript entitled “Evaluation of Aspergillus flavus Growth on Weathered HDPE Plastics Contaminated with Diesel Fuel.” Your insightful comments and suggestions have been invaluable in improving the clarity, rigor, and overall quality of our work.

In response to your review, we have made substantial revisions to the manuscript and addressed each of your observations point by point. For your convenience, we are providing the following documents:

  • A revised version of the manuscript with all changes tracked in Microsoft Word format.
  • A detailed response letter outlining how each of your comments was addressed.

We hope that the revised version meets your expectations and we remain at your disposal for any further clarification you may require.

With appreciation and respect,
César Antonio Sáez Navarrete
On behalf of all co-authors

Comment 1 Reviewer 1: Introduction around lines 87-98: The introduction is well written, but it could benefit from a brief review of existing studies on plastic biodegradation by fungi, not just hydrocarbons

Response 1 Reviewer 1: A new paragraph has been included as a result of a review of indexed and up-to-date literature, as requested. Modifications have been made in the document using Word’s Track Changes function to facilitate review of the edits.

The paragraph added is as follows, and the corresponding bibliographic references have been included in the References section. Please note that the numbering of the references has been updated accordingly to reflect the insertion of the new citations: “Fungal biodegradation of high-density polyethylene (HDPE) has garnered increasing attention due to the pressing need for sustainable solutions to plastic pollution. Various fungal taxa have demonstrated notable potential in degrading HDPE under laboratory conditions. Among them, Aspergillus species, such as A. niger, A. flavus, A. terreus, and A. fumigatus, have been widely reported to colonize polyethylene surfaces and induce chemical and morphological changes consistent with biodegradation, including weight loss and oxidation of polymer chains [9] [22]. Penicillium species, notably P. citrinum, P. chrysogenum, and P. oxalicum, have also been identified as effective agents in HDPE degradation, capable of reducing polymer mass and inducing visible cracks and microstructural damage on plastic films [21]. Similarly, members of the genus Fusarium, such as F. oxysporum and F. solani, have exhibited enzymatic capabilities to attack polyethylene matrices, resulting in measurable mass loss and alterations in infrared spectra. Recent investigations have even isolated effective HDPE-degrading fungi from unique ecosystems, such as mangrove soils (Aspergillus niger, Penicillium citrinum) and insect gut microbiota (e.g., Cladosporium halotolerans from Galleria mellonella larvae), underscoring the adaptability and metabolic diversity of fungi in plastic degradation [9]. These fungi secrete a wide range of oxidative enzymes, including laccases, peroxidases, and cutinases, which facilitate the breakdown of recalcitrant polymers into metabolizable oligomers. Collectively, these findings highlight the potential of Aspergillus, Penicillium, Fusarium, and related genera in advancing fungal-based strategies for mitigating HDPE pollution.”

Added references:

[9] Di Napoli, M., Silvestri, B., Castagliuolo, G., Carpentieri, A., Luciani, G., Di Maro, A., ... & Varcamonti, M. (2023). High density polyethylene (HDPE) biodegradation by the fungus Cladosporium halotolerans. FEMS Microbiology Ecology, 99(2), fiac148. https://doi.org/10.1093/femsec/fiac148

[21] Ong, G. H., Liew, L. M., Wong, K. K., Wong, R. R., Barasarathi, J., Loh, K. E., & Tanee, T. (2024). Screening of native fungi for biodegradation of high-density polyethylene (HDPE) plastic in mangroves ecosystem. Malaysian Applied Biology, 53(6), 97–103. https://doi.org/10.55230/mabjournal.v53i6.12

[22] Saira, A., Abdullah, L., Maroof, L., Iqbal, M., Farman, S., & Faisal, S. (2022). Biodegradation of low-density polyethylene (LDPE) bags by fungi isolated from waste disposal soil. Applied and Environmental Soil Science, 2022, 8286344. https://doi.org/10.1155/2022/8286344

Comment 2.1 Reviewer 1: Material and Methods: How were the replicates distributed across the experimental treatments? How was relative humidity controlled and monitored during incubation?

Response 2.1 Reviewer 1: The wording of Section 2.1 of the methodology has been improved to clarify the points raised by the reviewer, as follows:

“2.1. Study Objective and Experimental Design

This study aimed to evaluate the ability of the filamentous fungus Aspergillus flavus to colonize and grow on high-density polyethylene (HDPE) surfaces contaminated with diesel fuel. Three contamination conditions were considered: HDPE from a tank exposed to fresh diesel for less than two months, HDPE from a tank exposed to weathered diesel residues for over two years, and HDPE from a new, unused tank serving as an uncontaminated control. For each condition, six HDPE sections were collected and fabricated into 18 circular plastic discs, allowing for triplicate assays under each experimental setting. The samples were incubated at three distinct temperatures (20 °C, 25 °C, and 30 °C) under controlled environmental conditions, with relative humidity maintained at approximately 90%.

Fungal growth was monitored to assess the influence of temperature on colonization. Radial growth rate (RGR) was used as a kinetic metric to evaluate fungal expansion across HDPE surfaces, following established methodologies [15][16].”

Comment 2.2 Reviewer 1: Was the effectiveness of the UV decontamination treatment on plastic discs verified?

Response 2.2 Reviewer 1:

At the end of section 2.3. Preparation of HDPE Samples, the following paragraph has been added: “The effectiveness of UV-C decontamination (254 nm, 30 minutes per side) was verified by incubating a subset of sterilized HDPE discs on PDA without inoculation for 7 days at 25 °C. No microbial growth was observed, confirming the efficacy of the decontamination procedure.”

Comment 2.3 Reviewer 1: How was the application of PDA medium on the plastic discs standardized?

Response 2.3 Reviewer 1:

The original text…: “Once cooled to 50 °C, the medium was applied with a sterile brush onto the surface of the plastic discs, ensuring a thin and homogeneous coating.”

Has been replaced with: “Once cooled to 50 °C, the PDA medium was applied to the surface of the HDPE discs using a sterile brush, ensuring a thin and homogeneous coating. To standardize the volume, 100 µL of PDA was also applied per disc using a calibrated micropipette.” The medium was then allowed to solidify under sterile conditions, forming a consistent surface layer prior to fungal inoculation.”

Comment 2.4 Reviewer 1:  What specific identifier does the Aspergillus flavus strain used in the study have?

Response 2.4 Reviewer 1:

In Section 2.5. Inoculation and Incubation of the document, it is stated that: “The genetic sequence of the strain employed in this study has been published in [6].”

Citation 6 refers to a previous work by our group in which the genetic identification of the Aspergillus flavus strain used in this study was conducted, as indicated in the referenced source:

Cáceres-Zambrano, J.Z., Rodríguez-Córdova, L.A., Sáez-Navarrete, C.A. et al. Biodegradation capabilities of filamentous fungi in high-concentration heavy crude oil environments. Arch Microbiol 206, 123 (2024). https://doi.org/10.1007/s00203-024-03835-6.

Comment 2.5 Reviewer 1: Aspergillus flavus is known for producing aflatoxins, which are toxic and carcinogenic compounds. It is unclear whether the presence or the risk of aflatoxin production was evaluated in this study.

Response 2.5 Reviewer 1:

In Section 2.5. Inoculation and Incubation, at the end of the second paragraph, it is explicitly stated “The presence of aflatoxins was not evaluated in this study.”

Comment 3 Reviewer 1: Results: In the sections comparing fungal behavior on different types of plastic matrices and at various temperatures, mean values, standard deviations, and, in some cases, coefficients of variation are reported. However, it is not mentioned whether the observed differences between groups were subjected to statistical significance testing (e.g., ANOVA or t-test).

Response 3 Reviewer 1:

A new section has been created titled: “3.3.7. Influence of Temperature and Substrate Type on Fungal Growth: ANOVA Re-sults

At 30 °C, the analysis of variance (ANOVA) revealed statistically significant differences in the colony growth area of Aspergillus flavus among the different types of HDPE evaluated (contaminated with fresh diesel, weathered diesel, and clean control). The obtained p-value was 0.0018, well below the conventional threshold of 0.05, indicating that the observed differences are unlikely to be due to random variation. Furthermore, the F-statistic of 8.09 supports the presence of a substantial effect of substrate type on fungal development under elevated thermal conditions. These results suggest that, at 30 °C, the nature of the contaminant present on the HDPE surface plays a decisive role in the fungus's colonization capacity.

As temperature increased from 20 °C to 25 °C and subsequently to 30 °C, the differences in colony growth area of Aspergillus flavus among the various HDPE types became progressively more statistically significant. At 20 °C, no significant differences were detected between the substrate conditions. However, at 25 °C, a near-significant trend emerged, and by 30 °C, the differences were clearly significant, indicating a temperature-dependent enhancement of the fungus's discriminatory growth response to the physicochemical characteristics of the HDPE surfaces.”

The graphs showing the temporal evolution of colonies and halos (Figures 2-4) are very crowded, which makes interpretation difficult.

We appreciate the reviewer’s comment regarding the visual density of Figures 2–4. These graphs were intentionally designed to present the complete temporal dynamics of fungal growth and halo formation across all experimental conditions in a unified view. While we acknowledge that this comprehensive format may appear crowded, it allows for direct, side-by-side comparison of fungal behavior over time. To support interpretation, we have ensured that the legends, line styles, and markers are clearly differentiated, and the corresponding data are also provided in tabular format. We believe this approach strikes a balance between visual synthesis and informational completeness, and we hope the reviewer finds it acceptable.

Comment 4 Reviewer 1: Conclusions: There is no reflection on the experimental limitations, such as sample size, humidity control, or the extrapolation of results to real-world environments.

Response 4 Reviewer 1:

The following paragraph has been added at the end of the Conclusions section: “Despite the controlled nature of this study, certain limitations should be considered when interpreting the results. The sample size, while consistent with preliminary biodegradation studies, may limit the statistical resolution of comparisons across treatments. Although temperature and humidity were regulated using an integrated control system within the incubator, minor fluctuations may still occur and should be acknowledged. Moreover, while the use of HDPE samples contaminated under real-world weathering conditions adds ecological realism, extrapolating laboratory outcomes to field environments remains challenging due to the complexity of biotic and abiotic interactions. Future research should explore these dynamics in situ or under semi-controlled outdoor conditions”

Reviewer 2 Report

Comments and Suggestions for Authors

The topic is important and some new results have been obtained, while data on biodegradation of diesel fuel and plastic biodeterioration by fungi would be useful. The manuscript contains too many tables (some information is duplicated by figures), which can be converted into figures and combined.

Author Response

Dear Reviewer 2,

We sincerely thank you for your careful and constructive evaluation of our manuscript entitled “Evaluation of Aspergillus flavus Growth on Weathered HDPE Plastics Contaminated with Diesel Fuel.” Your insightful comments and suggestions have been invaluable in improving the clarity, rigor, and overall quality of our work.

In response to your review, we have made substantial revisions to the manuscript and addressed each of your observations point by point. For your convenience, we are providing the following documents:

  • A revised version of the manuscript with all changes tracked in Microsoft Word format.
  • A detailed response letter outlining how each of your comments was addressed.

We hope that the revised version meets your expectations and we remain at your disposal for any further clarification you may require.

With appreciation and respect,
César Antonio Sáez Navarrete
On behalf of all co-authors

Comment 1 Reviewer 2: The topic is important and some new results have been obtained, while data on biodegradation of diesel fuel and plastic biodeterioration by fungi would be useful. The manuscript contains too many tables (some information is duplicated by figures), which can be converted into figures and combined.

Response 1 Reviewer 2:

For greater clarity, we have presented the temporal evolution of fungal colony areas and the temporal evolution of halo areas in separate graphs for all tested temperatures (20 °C, 25 °C, and 30 °C), as follows:

Fig.4.

Fig. 3.

Fig. 2.

We thank the reviewer for the detailed suggestions regarding the organization of tables and figures. In response, significant adjustments have been made to enhance clarity and reduce redundancy:

Tables 1 and 2 have been merged into a single, more concise table.

Tables 3, 4, and 5 have also been combined and simplified into one unified table.

Similarly, Tables 6, 7, and 8, as well as Tables 9, 10, and 11, have each been consolidated into a single table per set, improving readability and alignment with the structure of the results.

In addition, Tables 13, 14, and 15 have been eliminated from the manuscript as recommended.

Although Figures 2, 3, and 4 were originally presented separately to aid clarity, we have considered the reviewer’s suggestion and ensured that their comparative interpretation is well supported in the text.

These modifications aim to streamline the presentation, avoid repetition, and facilitate easier navigation and comparison across experimental conditions.

Additional improvements

In addition to addressing your specific recommendations, we have also implemented several substantial improvements to the manuscript based on the feedback provided by Reviewer 1. Notably, the Introduction has been enriched with a new literature-based paragraph summarizing current research on the fungal biodegradation of HDPE by genera such as Aspergillus, Penicillium, and Fusarium, supported by updated references. Methodological sections were clarified to specify experimental replication, humidity control via an integrated incubator system, and the preparation and sterilization of HDPE discs. Details were also added regarding the PDA application standardization and strain identification, and a note was included to clarify that aflatoxin production by Aspergillus flavus was not evaluated in this study.

To enhance the rigor of the Results section, we incorporated a new subsection presenting ANOVA-based statistical analyses of fungal growth across HDPE conditions and incubation temperatures. These analyses revealed significant differences in colonization behavior, particularly at 30 °C. A brief discussion of experimental limitations—such as sample size, environmental control constraints, and the challenge of extrapolating laboratory data to real-world settings—was also added to the Conclusions. Together, these revisions substantially strengthen the clarity, robustness, and interpretability of the study.

Reviewer 3 Report

Comments and Suggestions for Authors

Dear authors,

The work assessed the ability of hydrocarbonoclastic A. flavus to grow on plastic surfaces and brought to light an important issue concerning the fate of plastic containers largely produced and accumulated in the environment. The manuscript also needs a comprehensive discussion section by listing concise references. The authors simply speculate. The topic of manuscript is deep and demand a report with much more than 24 references.  

Some important modifications must be made to enhance the quality of presentation. Here are my suggestions and coments:

  • Abstract: It is mandatory to italicize Aspergillus flavus (lines 19 and 32). Kindly check all text, as in A. flavus (line23); Penicillium (line 95) and lines 82, 117, 178, 195-196, 198, 227, and 303.
  • Note that only genera, species, must be italicized, so family Aspergillaceae (line 95) won’t be.
  • Kindly adopt “L” for liter (line 172), “min” for minutes (line 173
  • Section 2.1: what is the origin of isolate A. flavus? Recovery from environment by the authors? collection? Kindly provide this information and ID registration in case of a characterized strain.
  • Section 2.2.1: the term virgin may be replaced to fresh or commercial (note that is not a pure diesel. It contains additives, sulfur, water and impurities)
  • Adopt in the text that your weathered hydrocarbon samples aged 4 years (at least) and add the information in abstract which provided the information “more than 2 years”. See also line 279
  • As figure 1 summarizes tables 1 and 2, use one or other illustrative strategy. If you decide to maintain tables, I suggest presenting all data on a single table.
  • The same observation can be made to tables 3, 4 and 5. Plot all data on a single table. It helps compare variables.
  • On the affirmation in lines 281-283: the fact of larger colony area at 25º C may be caused by reduction of toxicity, however some information is missing. Was the strain psychrophilic? Was the strain recovered from the weathered area? To assume that toxicity is low is contradictory given information in lines 63-65, as well as heavy molecules are present which makes the contaminant persistent. Perhaps the authors should not speculate reasons in the results section.
  • Line 290: there is a scale in the literature to measure colony growth as weak, moderate and strong. Kindly provide it in the methodology section in order to use the term “moderate”.
  • On the presentation of tables 6, 7 and 8, see three suggestions above.
  • Do the same with tables 9, 10 and 11.
  • Remove table 13 and present only figure 2, same for table 14 and 15. I suggest showing figures 2, 3 and 4 as one. It is better to compare as well to follow the text.  
  • Compare the information of figure 3 and lines 281-283

Author Response

Dear Reviewer 3,

We sincerely thank you for your careful and constructive evaluation of our manuscript entitled “Evaluation of Aspergillus flavus Growth on Weathered HDPE Plastics Contaminated with Diesel Fuel.” Your insightful comments and suggestions have been invaluable in improving the clarity, rigor, and overall quality of our work.

In response to your review, we have made substantial revisions to the manuscript and addressed each of your observations point by point. For your convenience, we are providing the following documents:

  • A revised version of the manuscript with all changes tracked in Microsoft Word format.
  • A detailed response outlining how each of your comments was addressed.

We hope that the revised version meets your expectations and we remain at your disposal for any further clarification you may require.

With appreciation and respect,
César Antonio Sáez Navarrete
On behalf of all co-authors

Comment 1 Reviewer 3: Dear authors. The work assessed the ability of hydrocarbonoclastic A. flavus to grow on plastic surfaces and brought to light an important issue concerning the fate of plastic containers largely produced and accumulated in the environment. The manuscript also needs a comprehensive discussion section by listing concise references. The authors simply speculate. The topic of manuscript is deep and demand a report with much more than 24 references. 

Response 1 Reviewer 3:

A new paragraph has been added to the Introduction section based on a review of current, peer-reviewed literature addressing recent studies on plastic biodegradation by fungi. This addition was made to complement the previous version of the introduction, which was predominantly focused on literature related to hydrocarbon degradation. Accordingly, several new bibliographic references have been incorporated into the manuscript to support this expanded context.

“Fungal biodegradation of high-density polyethylene (HDPE) has garnered increasing attention due to the pressing need for sustainable solutions to plastic pollution. Various fungal taxa have demonstrated notable potential in degrading HDPE under laboratory conditions. Among them, Aspergillus species, such as A. niger, A. flavus, A. terreus, and A. fumigatus, have been widely reported to colonize polyethylene surfaces and induce chemical and morphological changes consistent with biodegradation, including weight loss and oxidation of polymer chains [9] [22]. Penicillium species, notably P. citrinum, P. chrysogenum, and P. oxalicum, have also been identified as effective agents in HDPE degradation, capable of reducing polymer mass and inducing visible cracks and microstructural damage on plastic films [21]. Similarly, members of the genus Fusarium, such as F. oxysporum and F. solani, have exhibited enzymatic capabilities to attack polyethylene matrices, resulting in measurable mass loss and alterations in infrared spectra. Recent investigations have even isolated effective HDPE-degrading fungi from unique ecosystems, such as mangrove soils (Aspergillus niger, Penicillium citrinum) and insect gut microbiota (e.g., Cladosporium halotolerans from Galleria mellonella larvae), underscoring the adaptability and metabolic diversity of fungi in plastic degradation [9]. These fungi secrete a wide range of oxidative enzymes, including laccases, peroxidases, and cutinases, which facilitate the breakdown of recalcitrant polymers into metabolizable oligomers. Collectively, these findings highlight the potential of Aspergillus, Penicillium, Fusarium, and related genera in advancing fungal-based strategies for mitigating HDPE pollution.”

Comment 2 Reviewer 3: Some important modifications must be made to enhance the quality of presentation. Here are my suggestions and coments:

Abstract: It is mandatory to italicize Aspergillus flavus (lines 19 and 32). Kindly check all text, as in A. flavus (line23); Penicillium (line 95) and lines 82, 117, 178, 195-196, 198, 227, and 303.

Note that only genera, species, must be italicized, so family Aspergillaceae (line 95) won’t be.

Response 2 Reviewer 3:

We thank the reviewer for this important observation. In response, we have italicized all instances of genus and species names, including Aspergillus flavus, A. flavus, and Penicillium, in the abstract and throughout the manuscript. We have also ensured that higher taxonomic ranks, such as the family name Aspergillaceae, remain in standard font as per taxonomic conventions. A thorough review of the entire document was performed to ensure consistency and compliance with scientific formatting standards.

Comment 3 Reviewer 3: Kindly adopt “L” for liter (line 172), “min” for minutes (line 173)

Response  Reviewer 3:

We appreciate the reviewer’s attention to unit consistency. The suggested changes have been implemented: “L” is now used for liter and “min” for minutes in the relevant sections. The manuscript has also been reviewed to ensure uniformity of these units throughout the text.

Comment 4 Reviewer 3: Section 2.1: what is the origin of isolate A. flavus? Recovery from environment by the authors? collection? Kindly provide this information and ID registration in case of a characterized strain.

Response 4 Reviewer 3:

In Section 2.5. Inoculation and Incubation of the document, it is stated that: “The genetic sequence of the strain employed in this study has been published in [6].”

Citation 6 refers to a previous work by our group in which the genetic identification of the Aspergillus flavus strain used in this study was conducted, as indicated in the referenced source:

Cáceres-Zambrano, J.Z., Rodríguez-Córdova, L.A., Sáez-Navarrete, C.A. et al. Biodegradation capabilities of filamentous fungi in high-concentration heavy crude oil environments. Arch Microbiol 206, 123 (2024). https://doi.org/10.1007/s00203-024-03835-6.

Comment 5 Reviewer 3: Section 2.2.1: the term virgin may be replaced to fresh or commercial (note that is not a pure diesel. It contains additives, sulfur, water and impurities)

Response 5 Reviewer 3:

We thank the reviewer for this clarification. As suggested, the term “virgin diesel” has been replaced with “commercial diesel” to more accurately reflect the composition of the fuel used, which includes common additives and trace components such as sulfur and water. This change has been applied in Section 2.2.1 and consistently throughout the manuscript where relevant.

Comment 6 Reviewer 3: Adopt in the text that your weathered hydrocarbon samples aged 4 years (at least) and add the information in abstract which provided the information “more than 2 years”. See also line 279

Response 6 Reviewer 3:

The manuscript has been revised to ensure consistency in the description of the plastic samples used in the study. Specifically, we now refer uniformly to HDPE tank samples exposed to environmental conditions for over two years as “weathered,” and those exposed for less than two months as “non-weathered.” This terminology has been applied consistently throughout the document to improve clarity and alignment with the experimental design.

Comment 7 Reviewer 3: As figure 1 summarizes tables 1 and 2, use one or other illustrative strategy. If you decide to maintain tables, I suggest presenting all data on a single table.

The same observation can be made to tables 3, 4 and 5. Plot all data on a single table. It helps compare variables.

Response 7 Reviewer 3:

The total number of tables in the manuscript has been reduced by six in accordance with the reviewer’s recommendation. The remaining tables have been properly renumbered, and all corresponding in-text citations have been updated to ensure consistency and coherence throughout the document.

Comment 8 Reviewer 3: On the affirmation in lines 281-283: the fact of larger colony area at 25º C may be caused by reduction of toxicity, however some information is missing. Was the strain psychrophilic? Was the strain recovered from the weathered area? To assume that toxicity is low is contradictory given information in lines 63-65, as well as heavy molecules are present which makes the contaminant persistent. Perhaps the authors should not speculate reasons in the results section.

Response 8 Reviewer 3:

The Aspergillus flavus strain used in this study was not characterized as psychrophilic, nor was it isolated from a contaminated environment, which limits our ability to draw definitive conclusions about its adaptive responses. In response to this valuable feedback, we have revised the sentence in question to remove speculative language and have relocated the broader discussion of potential influences on fungal growth to the Discussion section, where such interpretations are more appropriate. We have also ensured consistency with the chemical characterization presented earlier in the manuscript.

Comment 9 Reviewer 3:

Line 290: there is a scale in the literature to measure colony growth as weak, moderate and strong. Kindly provide it in the methodology section in order to use the term “moderate”.

Response 9 Reviewer 3:

he term “moderate” has been used in the manuscript in association with quantitative elements, in order to avoid confusion with other descriptors related, for instance, to fungal colony growth areas or halo expansion in the cultures.

Comment 10 Reviewer 3: On the presentation of tables 6, 7 and 8, see three suggestions above.

Do the same with tables 9, 10 and 11.

Remove table 13 and present only figure 2, same for table 14 and 15. I suggest showing figures 2, 3 and 4 as one. It is better to compare as well to follow the text. 

Response 10 Reviewer 3:

We thank the reviewer for the detailed suggestions regarding the organization of tables and figures. In response, significant adjustments have been made to enhance clarity and reduce redundancy:

Tables 1 and 2 have been merged into a single, more concise table.

Tables 3, 4, and 5 have also been combined and simplified into one unified table.

Similarly, Tables 6, 7, and 8, as well as Tables 9, 10, and 11, have each been consolidated into a single table per set, improving readability and alignment with the structure of the results.

In addition, Tables 13, 14, and 15 have been eliminated from the manuscript as recommended.

Although Figures 2, 3, and 4 were originally presented separately to aid clarity, we have considered the reviewer’s suggestion and ensured that their comparative interpretation is well supported in the text.

These modifications aim to streamline the presentation, avoid repetition, and facilitate easier navigation and comparison across experimental conditions.

Round 2

Reviewer 1 Report

Comments and Suggestions for Authors

Journal: Microorganisms (ISSN 2076-2607)

Manuscript ID: microorganisms-3659489

Type: Article

Title: Evaluation of Aspergillus flavus Growth on Weathered HDPE Plastics Contaminated with Diesel Fuel

The authors have addressed all the comments and recommendations from the initial review comprehensively. The revised manuscript now includes expanded background on fungal plastic degradation, clarifications regarding experimental design, and essential safety considerations related to Aspergillus flavus. Statistical analyses have been added where needed, and figures have been improved for clarity. Limitations are now acknowledged in the discussion.

Given these improvements, I consider the current version of the manuscript scientifically sound and suitable for publication.

Best regards.

Reviewer 3 Report

Comments and Suggestions for Authors

Dear César,

Thank you for yor response. I agree with your observations and thank you for accepting the suggestions. The manuscript is suitable for publishing.

All the best